# Changes in Fat Mass Following Creatine Supplementation and Resistance Training in Adults ≥50 Years of Age: A Meta-Analysis

**DOI:** 10.3390/jfmk4030062

**Published:** 2019-08-23

**Authors:** Scott C. Forbes, Darren G. Candow, Joel R. Krentz, Michael D. Roberts, Kaelin C. Young

**Affiliations:** 1Department of Physical Education, Faculty of Education, Brandon University, Brandon, MB R7A 6A9, Canada; 2Faculty of Kinesiology and Health Studies, University of Regina, Regina, SK S4S 0A2, Canada; 3School of Kinesiology, Auburn University, Auburn, AL 36849, USA; 4Department of Cell Biology and Physiology, Edward Via College of Osteopathic Medicine-Auburn Campus, Auburn, AL 36849, USA

**Keywords:** body composition, nutrition, supplement

## Abstract

Aging is associated with an increase in fat mass which increases the risk for disease, morbidity and premature mortality. Creatine supplementation in combination with resistance training has been shown to increase lean tissue mass in adults ≥50 years of age; however, the synergetic effects of creatine and resistance training on fat mass in this population are unclear. Creatine metabolism plays an important role in adipose tissue bioenergetics and energy expenditure. Thus, the combination of creatine supplementation and resistance training may decrease fat mass more than resistance training alone. The purpose of this review is two-fold: (1) to perform meta-analyses on studies involving creatine supplementation during resistance training on fat mass in adults ≥50 years of age, and (2) to discuss possible mechanistic actions of creatine on reducing fat mass. Nineteen studies were included in our meta-analysis with 609 participants. Results from the meta-analyses showed that adults ≥50 years of age who supplemented with creatine during resistance training experienced a greater reduction in body fat percentage (0.55%, *p* = 0.04) compared to those on placebo during resistance training. Despite no statistical difference (*p* = 0.13), adults supplementing with creatine lost ~0.5 kg more fat mass compared to those on placebo. Interestingly, there are studies which have linked mechanism(s) explaining how creatine may influence fat mass, and these data are also discussed.

## 1. Introduction

The prevalence of obesity (body mass index ≥30 kg/m^2^) in aging adults has risen substantially over the past few decades, with the highest rates in adults 60–75 years of age [1]. Specifically, intramuscular and visceral fat typically increase with minimal change in subcutaneous fat [1]. The age-related increase in fat mass is associated with greater risk for disease (i.e., cardiovascular disease, type II diabetes, obesity), morbidity and premature mortality [2]. Although multifactorial, one possible contributing factor explaining the age-related increase in fat mass is sarcopenia. Sarcopenia, defined as the age-related reduction in skeletal muscle mass, strength and physical performance [3], decreases physical activity, energy expenditure, and metabolic rate [4,5]. Therefore, lifestyle interventions which help overcome sarcopenia may also have a favorable effect on reducing fat mass in aging adults [6,7].

Resistance training is an effective lifestyle intervention for increasing muscle mass and strength in aging adults (for reviews see Hart and Buck [8]; Lopez et al. [9]). Additionally, resistance training increases participation in physical activity and activities of daily living leading to increased energy expenditure, metabolic rate and fat loss [10]. Combined with resistance training, dietary supplementation with creatine, a nitrogenous organic acid derived from reactions involving arginine, glycine and methionine in the liver and kidney [11], has beneficial effects on aging muscle. Three meta-analyses have been performed involving creatine supplementation during resistance training in older adults [12,13,14]. Collectively, results showed that the addition of creatine to resistance training significantly increased muscle mass (~1.21 kg), possibly by influencing high-energy phosphate metabolism, muscle protein kinetics, inflammation and oxidative stress, satellite cell activity and the expression of growth factors (for review see Candow et al. [15]).

Although most individuals supplement with creatine to increase muscle mass and/or muscle performance, rodent and in vitro studies provide mechanistic evidence suggesting that creatine also influences fat bioenergetics and energy expenditure [16,17,18,19]. Therefore, a review focusing on fat mass is warranted despite several recent reviews on creatine and resistance training in older adults that have focused on muscle mass and strength. Furthermore, individual human studies examining creatine supplementation on fat mass are mixed, with some showing a decrease in fat mass whereas others have shown no effect (Appendix A). Since the variability in the responsiveness to creatine supplementation is high in adults ≥50 years of age, and individual studies typically lack statistical power for detecting significant differences, it is important to determine with greater certainty whether creatine influences fat mass in adults ≥50 years of age. Therefore, our purpose was two-fold: (1) to perform meta-analyses on studies involving creatine supplementation during resistance training on fat mass in aging adults, and (2) to discuss possible mechanistic actions of creatine on reducing adiposity. This review is novel since, to our knowledge, it is the first to examine the impact of creatine on fat mass in aging adults.

## 2. Materials and Methods

Articles were selected for full-text review if they met the following criteria: (i) randomized controlled trial where participants were allocated to a creatine (≥2 grams per day) or a placebo treatment condition and resistance training; (ii) the mean age was ≥50 years; (iii) intervention duration ≥five-weeks of ≥twice per week of resistance training (to provide sufficient time and stimulus to demonstrate observable changes in fat mass); and (iv) include either absolute fat mass (kg), body fat percentage (%), or body mass changes (kg) over time. We included studies that combined creatine with other nutritional supplements (e.g., whey protein), but also ran our meta-analyses without these studies to determine whether they influenced the final outcome measures. If the outcome data was not able to be extracted, authors were contacted for missing information. Data extracted included pre- and post-training means and standard deviations (SD) or change scores for outcome variables and SDs for the change scores. When pre- and post-training means were extracted, change scores were calculated as post-training mean subtract pre-training mean. SDs for the change scores were estimated from pre- and post-training SDs (SDpre and SDpost) using the following equation derived from the Cochrane Handbook for Systematic Reviews of Interventions:SD change score = [(SDpre)^2^ + (SDpost)^2^ − 2 × (correlation between pre- and post-scores) × SDpre × SDpost]^1/2^(1)

In this equation, we used 0.8 as the assumed correlation between pre- and post-scores. Meta-analyses were run using RevMan 5.3 software (Cochrane Community, London, UK).

A fixed effect model was used. Weighted mean difference was calculated for all outcome variables, along with the 95% confidence interval (CI). Forest plots were generated for study-specific effect sizes along with 95% CIs and pooled effects. A p-value ≤ 0.05 was considered statistically significant. Statistical analysis for heterogeneity was assessed using chi-square test for heterogeneity and the I^2^ statistic to quantify total variation across studies attributable to heterogeneity. Funnel plots were generated and visually inspected for publication bias.

## 3. Results

Nineteen articles met our inclusion criteria (*n* = 609 participants). Individual study characteristics are shown in Appendix A. Fifteen studies used dual energy x-ray absorptiometry (DEXA), whereas other studies used bioelectrical impedance [20], skinfolds [21], hydrostatic weighing [22], and air displacement plethysmography [23] to determine changes in body composition. Resistance training duration ranged from 7 to 52 weeks (2 to 3 times per week). Creatine supplementation protocols used both relative (to body mass) and absolute dosing strategies with and without a loading phase (e.g., 20 grams for five to seven days). Training volume increased with creatine supplementation compared to placebo in two studies [24,25], while several others reported no differences between groups [22,23,26,27,28,29,30]. In addition, dietary intake (energy intake or protein) did not differ between groups [21,22,23,24,26,27,28,29,30,31,32,33,34,35,36,37,38].

Mean changes and SD for mean changes for individual studies, and pooled effects and their 95% Cis are presented along with Forest plots in Figure 1A–C. When pooling the data, body fat percentage (mean difference = −0.55% (95% CI = −1.08, −0.03); *p* = 0.04) decreased to a greater extent after creatine supplementation compared to placebo. Absolute fat mass (mean difference = −0.50 kg (95% CI = −1.15, 0.15); *p* = 0.13) and body mass (mean difference = 0.86 kg [95% CI = −0.32, 2.05]; *p* = 0.15) were not statistically significant between conditions. To explore confounding factors, meta-analyses were re-analyzed with (1) male or female only studies, (2) DEXA only studies, or (3) removal of multi-ingredient supplement studies. Final meta-analyses remained unchanged. There was no evidence of publication bias from the funnel plots.

## 4. Discussion

Results from this meta-analysis showed that adults ≥ 50 years of age who supplemented with creatine during resistance training experienced a greater reduction in body fat percentage compared to placebo (0.55%, *p* = 0.04; Figure 1A). Despite no statistical difference (*p* = 0.13), adults supplementing with creatine lost ~0.5 kg more fat mass compared to those receiving placebo (Figure 1B). We propose that the decrease in body fat percentage and fat mass from creatine supplementation may be clinically relevant as the age-related accumulation in fat mass increases the risk for disease (i.e., cardiovascular, type II diabetes, obesity), morbidity and premature mortality [2].

Mechanistically, there is direct evidence that creatine influences certain aspects of adipocyte and fat tissue metabolism as well as triglyceride synthesis in multiple cell types. In an elegant study, Kazak et al. [18] crossed mice which harbored a floxed allele of the creatine transporter with transgenic Adipoq-cre mice. The offspring lacked the creatine transporter in fat tissue (i.e., AdCrTKO mice) which resulted in significantly lower creatine and phosphocreatine levels. Relative to control mice, AdCrTKO mice presented a reduction in whole-body energy expenditure, a decrease in oxidative metabolism in beige and brown adipose tissue, and an increase in feed efficiency and whole body adiposity. Furthermore, these authors obtained adipose tissue from human subjects and demonstrated that the mRNA expression of the creatine transporter was negatively associated with BMI. In interpreting these findings, the authors suggested that creatine stimulates mitochondrial ATP turnover in fat tissue which, in turn, increases the metabolic rate of subcutaneous and brown adipose tissue. Notably, similar research has shown that a decrease in the biosynthesis of creatine in the fat tissue of mice blunts whole-body energy expenditure and increases fat accumulation [17,19,39]. Lee et al. [40] also reported that creatine inhibited the formation of cytoplasmic triglycerides in multiple adipogenic cell culture models in a dose-dependent manner through the inhibition of PI3K–Akt–PPARγ signaling. Earnest et al. [41] determined that 5 g/day of creatine supplementation in hypercholesteremic participants significantly reduced plasma triglyceride levels by ~20% after four to eight weeks of supplementation. While this effect was likely due to the inhibitory actions that creatine supplementation exerted on liver triglyceride synthesis, the collective evidence presented above suggests that the inhibitory effect that creatine has on triglyceride synthesis may also extend to adipose tissue. Results across in vitro and rodent studies indicate that creatine metabolism plays an important role in fat bioenergetics, and creatine supplementation positively influences energy expenditure. However, the effects of creatine supplementation on measures of lipolysis, energy expenditure and thermogenesis in aging adults remains to be determined.

Increased body mass (Figure 1C), combined with the decrease in fat mass and body fat percentage from creatine supplementation, may be indirectly related to and explained by greater muscle accretion from creatine supplementation and resistance training [12,13,14]. Greater lean tissue mass, which potentially increases resting metabolic rate and total daily energy expenditure (through increased participation in physical activities) [10,42] may possibly explain the decrease in fat mass and body fat percentage in aging adults supplementing with creatine. Furthermore, young males (21 ± 3 years of age) who supplemented with creatine (20 g/day for 5 days + 10 g/day for 23 days) during resistance training experienced a significant increase in lean tissue mass and resting metabolic rate/energy expenditure compared to resistance training and placebo [16].

## 5. Conclusions

In summary, this meta-analysis showed that creatine supplementation during resistance training has the potential to decrease body fat in adults ≥50 years of age. Decreasing fat mass is important for reducing the risk of disease (cardiovascular, type II diabetes, obesity), morbidity and premature mortality. While the mechanisms explaining the potential decrease in fat mass remain to be determined in humans, preliminary rodent data suggests creatine influences fat bioenergetics, metabolism, and energy expenditure.

## Figures and Tables

**Figure 1 jfmk-04-00062-f001:**
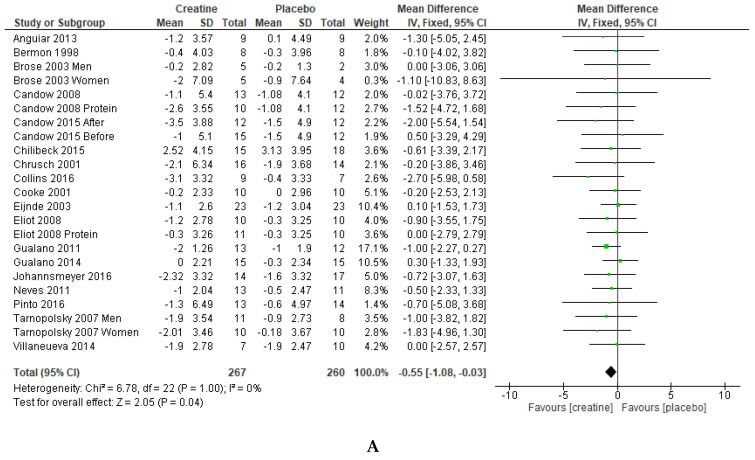
Forest plots for body fat percentage (%) (**A**), absolute fat mass (kg) (**B**) and body mass (kg) (**C**). Notes: Some studies presented data on men and women separately and on creatine and creatine + protein groups separately; therefore, these studies are entered twice in the meta-analysis for these separate subgroups. One study also presented data on participants who received creatine before versus after resistance training programs; therefore, these subgroups are entered separately in the meta-analysis.

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
