# Peer review of "Changes in Fat Mass Following Creatine Supplementation and Resistance Training in Adults ≥50 Years of Age: A Meta-Analysis"

_jfmk, 2019, doi:10.3390/jfmk4030062_

Round 1

Reviewer 1 Report

1.      It is needed commas between characters in Authors list section. Please add them.

2.      In Introduction (Ln 54), there is no Table 1 in the main text. Please check this.

3.      In Introduction (Ln 57), you use the expression “in adults ≥ 50 years of age” two lines before. I suggest replacing it with “in adults in this age range or in aging adults”.

4.      In Methods, the amount of the creatine supplementation and the protocol are important to induce changes, why not include a specific criterion for that?

5.      You included in your analysis 9 studies which did not inform about the training volume used in their interventions. Do you think that the volume of the training does not exert differences on found results? In addition, you included some studies which had different training volumes for CR group compared to placebo, do you think that it is appropriated to accept that kind of studies as methodologically correct?

6.      In your Methods (Ln 64), you must clarify the characteristics of the population, “i.e. healthy adults aged > 50 years aged”. In addition, after that clarification, you should include in your analysis just the population with those characteristics.

7.      In Methods (Ln 65), you used “five weeks and resistance trained” to define the type of intervention that you selected, you might use “five-week resistance training” to define it closely.

8.      On Ln 65, respect spaces between symbols and letter, please check it.

9.      On Ln 66, you used “percent body fat”. This variable appeared before as “body fat percentage”, also you used “body fat percentage” in the discussion. Please change it.

10.   How can you compare studies which use different tools to measure body composition variables?

I.E.: Do you think that the changes found by DEXA are similar to those found with bioelectric impedance?

Have you based the bias estimation on scientific evidence?

11.   Do you think that having or not a loading phase in the creatine supplementation cannot influence the possible effects of the creatine?

12.   On Ln 96, you used SD before. Please, be consistent with that and change it.

13.   You talk about supplementation but not about the control of diet, was that due to the lack of study data or because you didn't take it into consideration?

14.   On Ln 102, you used DEXA, but you have not used this abbreviation before, please add it in brackets in the right place.

15.   You must add the measure of each variable in Figure 1.

16.   In your Figure 1A and 1B, the placement of the “Favours creatine” and the one of “Favours placebo” are the same. However, in your Figure 1C you changed the order and it results confusing. Please, change it and put in the same order.

17.   In Table S1, the measurement of the duration of the training was weeks, please change in the second study (Bermon et al., 1998) “52 days” to weeks.

Author Response

Thank you for your suggestions. We have addressed all your concerns and revised the manuscript. All revisions are in yellow highlight.

1.        It is needed commas between characters in Authors list section. Please add them.

We have revised as suggested.

2.        In Introduction (Ln 54), there is no Table 1 in the main text. Please check this.

We have revised accordingly. We have changed it to “Table S1”. 

3.        In Introduction (Ln 57), you use the expression “in adults ≥ 50 years of age” two lines before. I suggest replacing it with “in adults in this age range or in aging adults”.

We have revised as suggested. “…in aging adults”

4.        In Methods, the amount of the creatine supplementation and the protocol are important to induce changes, why not include a specific criterion for that?

There are a variety of supplementation protocols that have been shown to be effective (e.g., Hultman et al., 1996, JAP). For example, 20 grams/day loading phase for 5 days followed by 2 grams per day was able to increase muscle creatine content ~20%. Similarly, 3 grams/day for 28 days increased muscle creatine content to a similar level (~20%).  We have included a statement suggesting that studies must have used ≥ 2 grams per day as per the literature. Furthermore, we wanted to be as inclusive as possible. We have done sub-group analysis to determine whether dosing strategy may have played a role; when only studies who did a loading phase were included, all the results were non-significant, this was because only 7 studies did a loading phase and thus substantially reduced the statistical power. 

5.        You included in your analysis 9 studies which did not inform about the training volume used in their interventions. Do you think that the volume of the training does not exert differences on found results? In addition, you included some studies which had different training volumes for CR group compared to placebo, do you think that it is appropriated to accept that kind of studies as methodologically correct?

Creatine can function through a variety of mechanisms. One of the proposed mechanisms is by enhancing training volume (i.e. creatine uptake into the muscle, which allows you do more reps and increase training volume). Thus we feel it is appropriate to include these studies. It was a surprise that some studies did NOT show an increase in training volume yet showed improvements in training adaptations. These studies suggest that creatine may work through other direct mechanisms in the muscle. We have recently published a review on the potential mechanisms involved in creatine (see. Candow et al., 2019; J Clin Med. 2019 Apr 11;8(4). pii: E488. doi: 10.3390/jcm8040488).

6.        In your Methods (Ln 64), you must clarify the characteristics of the population, “i.e. healthy adults aged > 50 years aged”. In addition, after that clarification, you should include in your analysis just the population with those characteristics.

We included both healthy populations and diseased state populations (one study used type 2 diabetics and one had participants with COPD). As long as the participants were able to complete the resistance training programs, we included the study. Again, we wanted to be as inclusive as possible. When we re-ran the meta-analysis percent body fat become non-significant when the Type 2 diabetes study was removed. This may be associated with a lack of statistical power. The heterogeneity I2 was 0%, suggesting that this study responded similarly to the other studies. 

    7. In Methods (Ln 65), you used “five weeks and resistance trained” to define the type of intervention that you selected, you might use “five-week resistance training” to define it closely.

We have revised as suggested.

8.        On Ln 65, respect spaces between symbols and letter, please check it.

We have checked the spaces and symbols. We hope it is satisfactory for the reviewer.

9.        On Ln 66, you used “percent body fat”. This variable appeared before as “body fat percentage”, also you used “body fat percentage” in the discussion. Please change it.

We have revised for consistency.

10.        How can you compare studies which use different tools to measure body composition variables?

We have performed a sub-analysis with only DEXA studies and the results of the meta-analysis were similar.  “To explore confounding factors, meta-analyses were re-analyzed with men or women only studies, or DEXA only studies, or the removal of multi-ingredient supplement studies, the final results of the meta-analyses remained unchanged.”

11. Do you think that the changes found by DEXA are similar to those found with bioelectric impedance?

We ran the sub-analysis and the meta-analysis results remained unchanged. As such, we can conclude that the inclusion of other tools (such as BIA) did not influence the results.

12. Have you based the bias estimation on scientific evidence?

We examined publication bias by visually inspecting the funnel plots based on criteria identified in the Cochrane Library handbook for systematic reviews.

13.     Do you think that having or not a loading phase in the creatine supplementation cannot influence the possible effects of the creatine?

We have re-ran the analysis with studies using creatine loading phase. Research has shown similar muscle creatine content following 28 days of low dose creatine supplementation protocol (3 grams per day) compared to the traditional 5 days of loading (20 grams per day) and then a maintenance phase (Hultman et al., 1996). The sub-analysis revealed that when only the loading studies were included, there was no significant effect of body fat percentage, however, this is due to the lack of statistical power, again since the heterogeneity was low.

14.     On Ln 96, you used SD before. Please, be consistent with that and change it.

We have revised as suggested.

15.     You talk about supplementation but not about the control of diet, was that due to the lack of study data or because you didn't take it into consideration?

The placebo used for most studies were maltodextrin. In table S1 we have reported on the dietary intake between control groups and creatine groups, most of which reported similar energy and protein intake between groups.  

16.     On Ln 102, you used DEXA, but you have not used this abbreviation before, please add it in brackets in the right place.

We have revised as suggested.

17.     You must add the measure of each variable in Figure 1.

We have revised as suggested.

18.     In your Figure 1A and 1B, the placement of the “Favours creatine” and the one of “Favours placebo” are the same. However, in your Figure 1C you changed the order and it results confusing. Please, change it and put in the same order.

We have revised accordingly.

19.   In Table S1, the measurement of the duration of the training was weeks, please change in the second study (Bermon et al., 1998) “52 days” to weeks.

We have revised and changed to "weeks".

Reviewer 2 Report

Body composition changes following creatine supplementation and resistance training in older adults is a valuable issue for age-related loss in muscle mass (sarcopenia). However, there are some systemic defect in this manuscript.

Previous meta-analyses have demonstrated that creatine supplementation during resistance training is effective for improving lean tissue mass and muscular strength (Candow et al., 2014; Chilibeck et al., 2017; Devries et al., 2014). This submitting manuscript perform meta-analyses on 19 articles (References No. 20-37, and Table 1; It seems only 18 articles??) from 2008 to 2016. As the manuscript describe “increased body mass, combined with the decrease in fat mass and body fat percentage from creatine supplementation, may be indirectly related and explained by the significant increase in lean tissue mass [12-14” (Candow et al., 2014; Chilibeck et al., 2017; Devries et al., 2014)] (in lines 131-133). In this submitting manuscript, the authors don’t discuss the difference between this work and the published papers. Actually, the references included in this submission manuscript and the selected criteria (ages of subjects, intervention of training, and body composition detection) of this manuscript are the same with Chilibeck et al. (2017). Study of Chilibeck et al. (2017) indicates that creatine supplementation resulted in greater increases in lean tissue mass (p<0.00001). This submitting manuscript showed that supplemented with creatine during resistance training experienced a greater reduction in body fat percentage (p = 0.04), and tended to favour absolute fat mass loss (p=0.13), and an increase in body mass (p=0.15). In my opinion, the term “tended to favour” express that the result is unconfirmed. I suggest that authors should discuss the differences result between this work and previously studies. Although the independent variables of Chilibeck et al. (2017) and this reviewing manuscript seems different, I think lean tissue mass is an important factor than body fat percentage for sarcopenia. Moreover, body mass include fat mass and fat free mass, and composition body fat percentage equal to fat mass divide to body mass. I doubt the result of reduction in body fat percentage but not on fat mass and body mass.

Therefore, the submission is poor in novelty. And I am sorry to say that I would recommend rejection of this manuscript.

Author Response

Thank you for your suggestions. We have addressed all your concerns and revised the manuscript. All revisions are in yellow highlight.

Body composition changes following creatine supplementation and resistance training in older adults is a valuable issue for age-related loss in muscle mass (sarcopenia). However, there are some systemic defect in this manuscript.

We agree that age-related changes in body composition are of importance. We have tried to address some of the defects and improve the clarity/focus of the manuscript.

Previous meta-analyses have demonstrated that creatine supplementation during resistance training is effective for improving lean tissue mass and muscular strength (Candow et al., 2014; Chilibeck et al., 2017; Devries et al., 2014). This submitting manuscript perform meta-analyses on 19 articles (References No. 20-37, and Table 1; It seems only 18 articles??) from 2008 to 2016.

There were 19 studies, we forgot to enter in one study into the table, which has now been corrected, as well as updating the reference list. Thank you very much for catching this error. 

As the manuscript describe “increased body mass, combined with the decrease in fat mass and body fat percentage from creatine supplementation, may be indirectly related and explained by the significant increase in lean tissue mass [12-14” (Candow et al., 2014; Chilibeck et al., 2017; Devries et al., 2014)] (in lines 131-133). In this submitting manuscript, the authors don’t discuss the difference between this work and the published papers. Actually, the references included in this submission manuscript and the selected criteria (ages of subjects, intervention of training, and body composition detection) of this manuscript are the same with Chilibeck et al. (2017).

We based our selection/inclusion criteria on Chilibeck 2017 as well as performing an updated search to identify any recent publications.  

Study of Chilibeck et al. (2017) indicates that creatine supplementation resulted in greater increases in lean tissue mass (p<0.00001).

This submitting manuscript showed that supplemented with creatine during resistance training experienced a greater reduction in body fat percentage (p = 0.04), and tended to favour absolute fat mass loss (p=0.13), and an increase in body mass (p=0.15). In my opinion, the term “tended to favour” express that the result is unconfirmed. I suggest that authors should discuss the differences result between this work and previously studies.

The previous meta-analyses were focused on lean tissue mass, however, with recent mechanistic rationale for creatine to enhance fat mass loss, it was logical to extend previous research, especially in older adults where fat mass gain is of concern. We have re-arranged the discussion to highlight the mechanistic work first and focus on the direct mechanisms whereby creatine may play a role in fat mass reduction. 

Although the independent variables of Chilibeck et al. (2017) and this reviewing manuscript seems different, I think lean tissue mass is an important factor than body fat percentage for sarcopenia.

We agree that lean tissue mass is important for sarcopenia; however body fat percentage is also an important health determinant. As the reviewer suggested, the outcome variables are different and thus we feel that this review is novel.

Moreover, body mass include fat mass and fat free mass, and composition body fat percentage equal to fat mass divide to body mass. I doubt the result of reduction in body fat percentage but not on fat mass and body mass.

We have made it clear throughout and in the conclusion that absolute fat mass were not statistically significant. 

Therefore, the submission is poor in novelty. And I am sorry to say that I would recommend rejection of this manuscript.

We feel that the manuscript is novel, since we examined different outcome variables. Our rationale is based on recent mechanistic research in rats suggesting a link between creatine and fat loss. As such, it is logical and rationale to examine the impact of creatine on fat mass, especially in older adults, where creeping obesity and sarcopenic obesity is of major health concern.

Round 2

Reviewer 1 Report

The article is well written, the study design is understandable and suitable to address the research questions and hypotheses of the authors. All in all, I would suggest accepting the manuscript for its scientific merits and accuracy.

Author Response

Point1 :The article is well written, the study design is understandable and suitable to address the research questions and hypotheses of the authors. All in all, I would suggest accepting the manuscript for its scientific merits and accuracy.

Response 1: Thank you very much for your time and suggestions.

Reviewer 2 Report

The authors corrected the cited articles of this meta-analyses and confirmed that they performing on 19 articles. However, the missing one published in 2008 (Eliot et al., No. 33) which also had been cited in a meta-analysis of Chilibeck et al. (2017). Although the authors replied that “We based our selection/inclusion criteria on Chilibeck 2017 as well as performing an updated search to identify any recent publications.” However, I think the main comment “the references included in this submission manuscript and the selected criteria (ages of subjects, intervention of training, and body composition detection) of this manuscript are the same with Chilibeck et al. (2017)” didn’t be replied. Moreover, in the section of introduction, the authors don’t discuss the difference between this work and the published papers of meta-analysis (references 12-15) in this revised manuscript. Therefore, the same opinion as last time, I think the submission is poor in novelty. And I am sorry to say that I would recommend rejection of this manuscript.

Author Response

Point 1: The authors corrected the cited articles of this meta-analyses and confirmed that they performing on 19 articles. However, the missing one published in 2008 (Eliot et al., No. 33) which also had been cited in a meta-analysis of Chilibeck et al. (2017). Although the authors replied that “We based our selection/inclusion criteria on Chilibeck 2017 as well as performing an updated search to identify any recent publications.” However, I think the main comment “the references included in this submission manuscript and the selected criteria (ages of subjects, intervention of training, and body composition detection) of this manuscript are the same with Chilibeck et al. (2017)” didn’t be replied.

Response 1: The Chilibeck 2017 ONLY examined changes in lean tissue mass and DID NOT examine changes in fat mass (which was the focus of this review). Here is the exert from the Chilibeck 2017 study: “The outcomes we assessed were whole-body lean tissue mass, determined with dual-energy X-ray absorptiometry, hydrostatic weighing, or air displacement plethysmography, and chest press and leg press muscular strength, representing global measures of upper and lower body strength, respectively”. We understand that the included studies were the same as Chilibeck 2017 since both reviews were interested in the effects of creatine and resistance training on body composition changes. Importantly, we extracted different outcome variables (fat mass vs. muscle mass), thus, making our review novel. I am unaware of any other review examining creatine and resistance training on fat mass in older adults.

Point 2: Moreover, in the section of introduction, the authors don’t discuss the difference between this work and the published papers of meta-analysis (references 12-15) in this revised manuscript.

Response 2: We have inserted a statement to make it clear how the reviews are different.

“Although most individuals supplement with creatine to increase muscle mass and/or muscle performance, rodent and in vitro studies provide mechanistic evidence suggesting that creatine also influences fat bioenergetics and energy expenditure [16-19]. Therefore, a review focusing on fat mass is warranted, despite several recent reviews on creatine and resistance training in older adults that have focused on muscle mass and strength.”

We also added in a statement to make it clear that this is the first review on creatine and resistance training on fat mass in aging adults.

“This review is novel since, to our knowledge, it is the first to examine the impact of creatine on fat mass in aging adults. “

Therefore, the same opinion as last time, I think the submission is poor in novelty. And I am sorry to say that I would recommend rejection of this manuscript.